# Alzheimer’s Disease and Green Tea: Epigallocatechin-3-Gallate as a Modulator of Inflammation and Oxidative Stress

**DOI:** 10.3390/antiox12071460

**Published:** 2023-07-20

**Authors:** Víctor Valverde-Salazar, Daniel Ruiz-Gabarre, Vega García-Escudero

**Affiliations:** 1Department of Anatomy, Histology and Neuroscience, School of Medicine, Universidad Autónoma de Madrid, 28029 Madrid, Spain; valverdesalazarv@gmail.com (V.V.-S.); d.ruizgabarre@gmail.com (D.R.-G.); 2Centro de Investigación Biomédica en Red de Enfermedades Neurodegenerativas, CIBERNED, 28031 Madrid, Spain; 3Institute for Molecular Biology-IUBM, Universidad Autónoma de Madrid, 28049 Madrid, Spain

**Keywords:** Alzheimer’s disease, EGCG, neuroprotection, antioxidant, green tea, amyloid β, tau

## Abstract

Alzheimer’s disease (AD) is the most common cause of dementia, characterised by a marked decline of both memory and cognition, along with pathophysiological hallmarks including amyloid beta peptide (Aβ) accumulation, tau protein hyperphosphorylation, neuronal loss and inflammation in the brain. Additionally, oxidative stress caused by an imbalance between free radicals and antioxidants is considered one of the main risk factors for AD, since it can result in protein, lipid and nucleic acid damage and exacerbate Aβ and tau pathology. To date, there is a lack of successful pharmacological approaches to cure or even ameliorate the terrible impact of this disease. Due to this, dietary compounds with antioxidative and anti-inflammatory properties acquire special relevance as potential therapeutic agents. In this context, green tea, and its main bioactive compound, epigallocatechin-3-gallate (EGCG), have been targeted as a plausible option for the modulation of AD. Specifically, EGCG acts as an antioxidant by regulating inflammatory processes involved in neurodegeneration such as ferroptosis and microglia-induced cytotoxicity and by inducing signalling pathways related to neuronal survival. Furthermore, it reduces tau hyperphosphorylation and aggregation and promotes the non-amyloidogenic route of APP processing, thus preventing the formation of Aβ and its subsequent accumulation. Taken together, these results suggest that EGCG may be a suitable candidate in the search for potential therapeutic compounds for neurodegenerative disorders involving inflammation and oxidative stress, including Alzheimer’s disease.

## 1. Introduction

Dementia consists of a neurological syndrome characterized by memory, behaviour and language impairment that progressively builds up, leading to an inability to perform activities of daily living [1,2]. Alzheimer’s disease (AD), an eminently multifactorial neurodegenerative disorder, is considered to be the most frequent cause of dementia [2,3]. AD mainly manifests through short-term memory deficits, along with other cognitive, affective, psychosocial and, less frequently, motor impairments [2,3,4,5,6].

Current frameworks establish that, temporally, pathophysiological evidence of AD can be found 20–30 years before the beginning of symptoms that constitute the clinical phase of the disease, with molecular, cellular, biochemical, and functional alterations that have been linked to the development of symptoms later in life [7,8].

The main histopathological hallmarks of the disease are the accumulation of amyloid beta (Aβ) in the form of extracellular plaques, the aggregation of hyperphosphorylated tau protein, conforming intracellular neurofibrillary tangles, as well as evidence of neurodegeneration associated with amyloid and tau pathologies forming neuritic plaques [9,10,11]. All these are accompanied by other factors, including inflammation that implies astrocyte and microglial involvement and less specific mechanisms, such as Lewy bodies and vascular alteration [12,13,14]. The multifactorial nature of AD, together with the diffuse timeline differentiating preclinical and clinical stages and the fact that some of the predominant signs of the disease are shared by other, unrelated disorders, constitute the main causes that explain the current lack of efficient treatments to cure, delay or palliate the effects of this terrible disease [4,7].

Faced with this disheartening lack of therapeutic resources, several authors have proposed alternative approaches to the modulation of Alzheimer’s disease and other neurodegenerative disorders, by intervening in environmental and lifestyle factors such as encouraging physical exercise, avoiding alcohol and drug consumption, lowering stress levels, maintaining cognitive stimulation or promoting a healthy, balanced diet [15,16,17,18,19]. 

In this line, dietary modulation has been promoted as one of the more accessible interventions to promote and maintain health during people’s lifespan [17,20,21]. As one of the most extensively consumed beverages, only second to water, and due to the bioactive nature of its components, tea has acquired some popularity in recent years as a source of potentially beneficial compounds to help tackle complex pathophysiological processes [22,23]. 

Tea is obtained from the leaves of Camellia sinensis, a plant species native to East Asia that has been consumed for more than 4000 years [23,24]. Different varieties of tea can be obtained from the desiccated leaves of the plant, depending on the processing [23,25]. Green tea is obtained by steaming and drying freshly harvested leaves, which grants a composition virtually identical to that of the leaves and a higher concentration of bioactive molecules with potential benefits [24,25]. However, if the leaves are left to undergo fermentation, we can obtain oolong tea (small fermentation window) and black tea (longer fermentation window) [23,25]. Fermentation entails the oxidation of around 20–30% of their flavonoids, which leads to the loss of certain bioactive molecules such as polyphenols [25].

Green tea comprises several bioactive compounds, such as polyphenols, caffeine and amino acids [22,24]. Green tea polyphenols, often referred to as catechins, amount to around 30% of the dry weight of the leaves and have been proposed to mediate several health benefits [22,24,25]. Epigallocatechin gallate (EGCG), an ester of gallic acid and epigallocatechin, is the most abundant catechin within green tea, constituting up to 65% of total catechin content and displaying the highest biological activity [22], including neuroprotective and antioxidative effects reported in the literature [22,23,26,27].

In accordance with this, several epidemiological studies in the Asian population have linked green tea consumption and a decreased risk for neurodegenerative disorders, such as AD, Parkinson’s disease and other dementias. The effects of EGCG and its abundance within green tea, together with the fact that it can be efficiently absorbed in the intestine [28,29], constitute two key points of the potential therapeutic use of EGCG. In addition, since AD pathophysiology occurs mainly in the brain, the ability to cross the blood–brain barrier is a necessary feature of any potential therapeutic agent. In this regard, EGCG has been proven to cross the blood–brain barrier even at very low concentrations [30,31,32,33], which positions this catechin as a potential mediator with beneficial properties for Alzheimer’s disease and other forms of neurodegeneration [22,26,27,34].

In the present work, we review the available evidence regarding the neuroprotective and neuroregenerative role of EGCG and the molecular mechanisms that may be involved in such an effect, contextualising the results within the pathological mechanisms that mediate neurodegeneration in Alzheimer’s disease. In consequence, we aim to propose EGCG and green tea as a potentially useful tool within dietary interventions in patients living with neurodegenerative, incurable diseases.

## 2. Molecular Mechanisms of Alzheimer’s Disease

Since Alzheimer’s disease was first described in 1906 [35], intensive research has led to the proposal of numerous working hypotheses with regard to the origins of the pathology and the molecular mechanisms involved in the process [5,9,36]. Classically, amyloid and tau have been considered to be the main actors of the disease, with the amyloid cascade hypothesis enjoying the most prestige during the end of the past century and the beginning of ours; albeit current approaches tend to favour different perspectives, including oxidative stress and inflammatory involvement or a systems biology point of view [5,37,38,39].

Due to this multifactorial nature, the analysis of the potential causes of Alzheimer’s merits a review of its own and would far exceed the scope of this work. Thus, we intend only to detail some of the molecular aspects of Alzheimer’s pathology to contextualise the potential benefits of EGCG as a modulator of these mechanisms. For a thorough review of Alzheimer’s pathology, causes and treatments, readers can resort to [5,40].

### 2.1. Amyloid Pathology

As mentioned, one of the hallmarks of Alzheimer’s disease is the accumulation of amyloid β in extracellular plaques throughout the brain of patients [41,42]. On a molecular level, amyloid pathology arises from the processing of the transmembrane protein amyloid precursor protein (APP) [43].

APP can suffer so-called amyloidogenic processing, upon sequential cleavage by the β-secretase BACE1 and a γ secretase [43,44]. The former produces a soluble fragment (sAPPβ) and a C-terminal one (CTFβ), while the latter uses this CTFβ as a substrate, liberating an intracellular domain into the cytoplasm and generating amyloid β (Aβ) fragments whose length depends on the exact point of cleavage, with Aβ40 and Aβ42 being the predominant species [40,43,44]. These Aβ monomers can then aggregate to different extents, giving raise to the mentioned plaques that would then mediate the pathological effects described in the disease [43,44], although evidence from the past decade points toward oligomeric forms as the main drivers of the disease [5,45] (Figure 1 (A)). 

However, most of the processing of APP in physiological conditions is directed through the non-amyloidogenic route, in which the first cleavage is carried out by an α-secretase, including several zinc metalloproteases belonging to the disintegrin and metalloproteinase family (ADAM) with ADAM10 being the main α-secretase in the human brain [44]. Such a cleavage produces a soluble fragment (sAPPα) and a C-terminal one (CTFα), also susceptible of being cleaved by a γ-secretase [43,44]. Importantly, sAPPα exerts neuroprotective functions, especially in the context of traumatic brain injury and neurodegeneration, but also acts in antagonizing the amyloidogenic route, by inhibiting BACE1 [38,43] (Figure 1 (B)).

### 2.2. Tau Pathology

Tau is an intrinsically disordered, highly soluble protein, and is involved in a great deal of physiological functions, such as microtubule assembly and stabilisation [46,47]. Tau functions are regulated, among other mechanisms, through phosphorylation, by means of a dynamic balance between kinases and phosphatases [11,48,49,50]. 

In pathological conditions, such as those of Alzheimer’s disease, the loss of this balance leads to the hyperphosphorylation of tau protein, reaching 3–4 fold levels of phosphorylation than in physiological conditions [11]. This results in an inability of tau to maintain its physiological functions, provoking microtubule destabilisation, and inducing sequential tau aggregation into oligomers and paired-helicoidal filaments that lead to the final formation of neurofibrillary tangles [11,51,52] (Figure 1 (8–10)). Again, more recent evidence suggests that neurofibrillary tangles themselves are not toxic, but it is the intermediate species being the ones that act as pathological drivers [53,54]. 

### 2.3. Other Drivers of Alzheimer’s Pathology

Amyloid and tau pathology are accompanied by neuronal death and neurodegeneration, completing the characteristic triad that constitutes what recent guidelines have termed the Amyloid–Tau–Neurodegeneration (ATN) axis [55]. We can observe this neurodegeneration on a tissular level in the form of neuritic plaques, where amyloid and tau pathology converge and dystrophic neurites can also be observed [56] (Figure 1 (1–4)).

However, the intensive rhythm of research around Alzheimer’s disease has allowed the discovery of other mechanisms involved in the pathology of AD, including neuroinflammation, oxidative stress, mitochondrial dysfunction, metal homeostasis, autophagy alterations, vascular implication, synaptic deficits, etc. [5,45,57,58,59,60]. Most of these mechanisms can coexist and actually do, establishing complex, often bidirectional relationships that make it virtually impossible to ascertain one single cause of the disease at this point. Some of these will be reviewed in the present work in the context of the neuroprotective functions of the green tea catechin EGCG.

## 3. Alzheimer’s Disease and Oxidative Stress

Redox balance is one of the most important mechanisms of homeostasis at a cellular level, with both free radicals and antioxidants exerting cell signalling and regulatory functions in physiological conditions [61,62,63]. 

In the context of neurodegenerative disorders, it is important to consider that the brain presents energy requirements well above the rest of the organs, which explains that it consumes approximately 20% of the oxygen obtained through the respiratory system [64,65]. As a consequence, brain cells are more likely to generate free radicals, such as reactive oxygen and nitrogen species (ROS and RNS, respectively) and therefore have to ensure antioxidant mechanisms can cope with these to provide an adequate redox balance [62,65]. In pathological conditions, including Alzheimer’s disease, this balance is altered, causing the oxidation of nucleic acids, lipids and proteins in neurons through several mechanisms [57,66,67]. For instance, neurons contain high amounts of polyunsaturated fatty acids that are able to react with ROS, which prompts a cascade of lipidic peroxidation that leads to cellular death [62,65,66]. These kinds of mechanisms are critical in neurons, where the levels of glutathione, one of the most important antioxidant compounds to deal with free radicals, are decreased [68,69], even more so in ageing and neurodegenerative diseases [63,68,70].

Evidence has been accumulating so as to consider AD as an oxidative stress disease, with an increased production of free radicals [63,67] and a reduction in antioxidant enzymes such as catalase or superoxide dismutase (SOD) [57,65], which aggravates the redox imbalance due to ageing and the vulnerability of neurons [58,70].

This evidence has allowed researchers to establish links between oxidative stress and a number of pathology-mediating mechanisms in AD [66,67,71,72]. More specifically, amyloid and tau pathology has been directly linked to oxidative stress in a bidirectional manner, with oxidative stress aggravating both pathologies [57,65,71] (Figure 2 (1,2)).

Proteomic studies show a relationship between tau pathology in vivo, in P301S and P301L mice that model tauopathies, and mitochondrial dysfunction and an NADH dehydrogenase decrease, which causes alterations in the respiratory chain and ATP synthesis [62,65,73,74]. In addition, in vivo experiments in the Drosophila melanogaster tauopathy model tau R406W showed a tau-induced reduction in antioxidative systems such as SOD and vitamins C and E, promoting histological alterations and apoptosis. In fact, the overexpression of such antioxidative systems was proven to be enough to counter tau-mediated neuronal death [75] (Figure 2 (1,7)).

In turn, oxidative stress contributes to tau pathology through diverse mechanisms. On the one hand, oxidative stress conditions favour tau hyperphosphorylation, one of the main drivers of disease, via the regulation of GSK3 and PPA2, the main kinase and phosphatase involved in tau phosphorylation, respectively [49,58,65,76]. On the other hand, oxidised fatty acids can induce tau polymerisation, through unclear pathways that seem to be cysteine-dependent [65,76]. 

As for amyloid β, the mechanisms that mediate the relationship between amyloid pathology and oxidative stress seem to be quantity-dependent. Low levels of amyloid peptide accumulation exert a self-protective effect via the inhibition of lipoprotein oxidation in CSF and plasma [65,77,78]. However, upon reaching a certain threshold of accumulation, amyloid deposits cause alterations in the respiratory chain mediated by mitochondrial and enzymatic dysfunction [79,80,81]. Indeed, in vitro studies prove that Aβ causes oxidative damage to the mitochondrial membrane, deteriorating the lipid polarity and hindering the transport of protein and enzymes required in the respiratory chain [81,82]. Such alterations hamper mitochondrial transport and increase oxidative stress, which in turn promotes mutations in mitochondrial DNA [79,83]. In fact, amyloid β has been reported to produce deficits in cytochrome c oxidase and ATP production, while increasing hydrogen peroxide and nitric oxide in vivo in transgenic Tg2576 mice, which induces protein and lipid oxidation, as well as apoptosis, even before the accumulation of amyloid plaques [84,85,86]. Additionally, in vivo evidence from amyloid mice model APP/PS1 showed SOD inactivation, which aggravated mitochondrial dysfunction, enhanced oxidative stress and promoted apoptosis [87]. Conversely, the overexpression of SOD in vivo in Tg19959 (APP_695_) mice resulted in a reduced amyloid plaque load, a diminished protein oxidation and memory restauration [88] (Figure 2 (2,5)). 

Directly related to tau and amyloid pathology, and calling back to the idea that neurofibrillary tangles and amyloid plaques themselves appear to be not as much of disease effectors as once thought, several authors propose that they may act as some sort of protective mechanisms with antioxidant capacities [41,49,70,72]. Indeed, the degree of oxidative stress correlates inversely with neurofibrillary tangle accumulation [49,58,70], possibly partly mediated by the fact that neurofibrillary tangles have proven to be able to sequester oxidised species by means of lysine–serine–proline domains within tau [49]. As for senile plaques, they have been demonstrated to bind ions that would otherwise promote oxidative stress if free [41,72]. 

In addition, both tau and amyloid pathology have been linked to neuroinflammation, as drivers of inflammation but also consequences of the inflammatory state [12,89]. Inflammation, whatever its origins, actively contributes to ROS production and neuronal damage, through the microglial and astrocytic release of proinflammatory substances such as cytokines, chemokines and complement proteins upon activation [12,40,90] (Figure 2 (4,8,9)).

Creating an even more complex picture in the interaction between AD and oxidative stress, mitochondrial dysfunction has been proposed to be one of the earliest manifestations of AD [62,72,91]. This is especially relevant in the context of oxidative stress, since mitochondria are the main producers of ROS [67,80]. As briefly mentioned, tau and amyloid pathology undoubtedly contribute to this dysfunction, for example, by inducing a deficit of NADH dehydrogenase or cytochrome oxidase [65,74,88] (Figure 2). 

Lastly, metal accumulation has been proposed as yet another common mechanism between oxidative stress and Alzheimer’s disease, with some authors suggesting that copper, iron and zinc accumulation in AD are the main drivers of oxidative stress associated with the pathology [61,62,67]. All these metals are able to bind amyloid β peptides and promote their aggregation, but copper forms the most stable complexes that lead to the generation of superoxide anion and hydrogen peroxide upon copper reduction [61,92] (Figure 2 (10–12)).

Iron has been equally linked to neurodegenerative diseases and AD in particular, promoting not only amyloid deposition but also tau phosphorylation and aggregation into neurofibrillary tangles [93]. Furthermore, an iron responsive element (IRE type II) has been described within the 5′UTR region of APP mRNA, which, upon activation by iron ions, is susceptible of enhancing endogenous APP translation, subsequently facilitating amyloid deposition [94,95]. Additionally, iron has been demonstrated to contribute to the production of ROS through the Fenton reaction that can lead not only to oxidative stress but also prompt an inflammatory cascade that stimulates the production of cytotoxic cytokines in microglia [94,96] (Figure 2).

Moreover, ferroptosis has been described as a type of programmed cell death caused by iron overload that leads to the failure of the glutathione-dependent antioxidant defences, resulting in the ROS overproduction and accumulation of lipid peroxides. During ferroptosis, neurons release lipid metabolites from inside the body that are harmful to the surrounding neurons, causing inflammation, and having significant implications in a haemorrhagic stroke, Alzheimer’s disease and Parkinson’s disease [97] (Figure 2). It has been suggested that cellular senescence in AD involving astrocytes, oligodendrocytes, microglia and even neurons increases iron content, potentially leading to neuronal cell death with ferroptosis and subsequent inflammation processes [98]. Indeed, magnetic resonance imaging has shown iron deposits in an AD brain as well as increased transferrin and ferritin, suggesting an augmented iron intake in these patients [99]. At the same time, the transporter that mediates the secretion of Fe^3+^ named ferroportin has been shown to be downregulated by Aβ, diminishing iron excretion [99,100]. Additionally, Aβ has been shown to decrease the levels of GPX4, an antioxidant enzyme that regenerates reduced glutathione, inhibiting ferroptosis [99]. Therefore, an AD brain is very susceptible to developing ferroptosis, pointing out the inhibition of this type of cell death as a therapeutic strategy for AD [98,99,101].

Zinc, however, seems to play a more complex role. While high concentrations of zinc have been reported to result in neuronal death independently or synergistically with amyloid deposition, micromolar concentrations of zinc demonstrated a protective effect against Aβ toxicity [61,102,103]. In this fashion, iron and specially zinc binding to amyloid peptides may constitute a protective mechanism by precluding copper binding, thus avoiding the production of oxidised species such as hydrogen peroxide [61].

## 4. Neuroprotective Effects of EGCG in the Context of Alzheimer’s Disease

### 4.1. Antioxidative Effects of EGCG

All the previous evidence allows us to conclude that there is a complex relationship between AD and oxidative stress that implies a positive feedback loop between both and poses oxidative stress and the mechanisms mentioned above as potential targets in our efforts to mitigate AD pathology. Thus, antioxidants such as green tea catechins could potentially offer interesting incorporations to our therapeutic arsenal. Assuredly, catechins have proven to be able to act as antioxidative systems, neutralising ROS and RNS and other free radicals, such as nitric oxide, peroxyl, peroxynitrite, carbon and lipidic radicals or 1,1-diphenyl-3-picrylhydrazine derivatives [94] (Figure 3 (8)). 

EGCG has been proven to exert a greater antioxidative effect than other green tea catechins such as epicatechin or epigallocatechin, greater even than that of potent antioxidants such as vitamins E and C [94,104,105], linked to its structure that leaves several hydroxyl groups free to sequester radicals (Figure 3 (6)).

In addition, EGCG and other flavonoids, together with phenolic antioxidants found in green tea, can also activate endogenous antioxidative systems, thus exerting an indirect protective effect [34,94]. Namely, flavonoids promote the expression of stress response genes, including those coding for the enzymes haem oxygenase and glutathione-S-transferase [106], by binding to antioxidant regulatory elements (ARE) in the promoter of said genes [34,106] (Figure 3 (1–3)). Furthermore, under oxidative stress conditions, the activation of these genes seems to be accompanied by the modulation of MAPK’s function, promoting the activity of transcription factors Nrf1 and Nrf2 and increasing their nuclear binding to ARE sequences [106,107].

Moreover, EGCG showed an antioxidative effect in vitro in neuronal primary cultures, where it inhibited the toxic effects of 3-hydroxykynurenine, a metabolite of tryptophan that acts as a potent endogenous neurotoxin, increasing oxidative stress and promoting ROS production [94,108]. Upon EGCG-mediated inhibition, oxidative stress and ROS were reduced and caspase activation and apoptosis were diminished consequently [108] (Figure 3 (8)). 

Synergically with all these effects, EGCG enhanced the activity of SOD and catalase, two of the most relevant endogenous antioxidative systems, which proved to be enough to reduce oxidative stress in C57BL mice [94,109] (Figure 3 (5)).

### 4.2. Iron-Chelating Effects of EGCG

The capability of EGCG and other catechins to bind metal ions poses another exciting overlap between oxidative stress, AD and the neuroprotective effects of catechins. More specifically, catechins have proven to be exceptional iron and copper chelators [94,110,111], which is especially relevant, given that iron metabolism alterations have been proposed to be a common link between several neurodegenerative disorders [112,113,114,115] (Figure 2 (16,17), Figure 3 (6,7) and Figure 4 (6,7)). 

Given the numerous pathways in which iron imbalance can modulate pathological mechanisms in Alzheimer’s disease, iron chelation promises an interesting research avenue to explore. In fact, iron chelators have been proposed as potential multi-target treatments for Alzheimer’s disease throughout the last decades [112,113,116]. In this regard, iron chelation exerts a direct neuroprotective effect in AD by avoiding the iron-mediated promotion of amyloid and tau pathology, through the mechanisms detailed above [93,95,96,116]. For instance, the use of iron chelators to assert a neuroprotective effect relies on both the elimination of excess iron in the brain and the prevention of its accumulation under oxidative stress conditions [116]. For instance, a prolonged administration of EGCG to C57BL mice in vivo was shown to diminish hippocampal APP without modifying APP mRNA, which suggests a post-transcriptional level of intervention, such as intracellular iron chelation [117] (Figure 4 (6,8,9)).

Apart from that, one of the main neuroprotective effects arising from iron chelation is associated with the activation of the hypoxia inducible factor 1α (HIF-1α) pathway that results in the stabilisation of the transcription factor HIF-1, involved in the transcription of cell survival and oxidative stress response genes [94,116,118]. A HIF-1α presence depends on the activity of HIF-prolyl-4-hydroxylases, which are iron-dependent enzymes. In the face of an overload of iron, these enzymes catalyse the hydroxylation of proline and asparagine residues within HIF that target its degradation via the proteasome. Thus, under excess iron conditions, those oxidative-stress-response and cell-survival genes’ expression is decreased [94,116]. On the contrary, EGCG has been proven to act as a direct HIF-1α activator, thus promoting cell survival genes and neuroprotection [34] (Figure 4 (7,15,16)).

Precisely, in relation to cell survival in neurons, intracellular iron modulation has been extensively proposed as a means to avoid apoptosis and stop the cell cycle [113,116,119]. In AD in particular, a dysregulation of the cell cycle has been equally characterised, with consequences such as cytoskeleton phosphorylation, mitochondrial abnormalities, and alteration in several transduction pathways, such as those of GSK3, CDK5 and ERK2 [119,120,121]. The activation of these pathways consequently produces aberrant tau phosphorylation, DNA replication and an increased expression of cell cycle proteins such as cyclins (A, B, D and E) [116,119]. Other reports show that control mechanisms on phases G1 and S of the cell cycle cannot be correctly performed in AD, allowing neurons to continue the cell cycle and progress to G2, even completing DNA replication, observing 3–4% of tetraploid cells [121,122]. In line with this, some genes related to the cell cycle have been proven to be altered under oxidative stress conditions such as the ones found in AD patients. For instance, PIN1, a gene with important implications in the correct regulation of the cell cycle, has been proven to regulate age-dependent neurodegenerative processes, including APP processing and tau dephosphorylation [123]. In the context of Alzheimer’s disease and oxidative stress, PIN1 can be inhibited, leading to either mitotic arrest or neuronal death [123]. In the same line, BRCA1, a gene with a role in cell growth and DNA repair, was overexpressed in neurons that possessed neurofibrillary tangles in AD, which would entail an increased genome instability [124].

Iron seems to carry out a regulatory function on the cell cycle, in that under iron deficiency conditions, cells cannot advance from the G1 phase to S, a regulatory role directly related to D1 cyclin degradation [125], while iron accumulation has been reported to disrupt the cell cycle, promoting abnormal progression through the cycle that leads to apoptosis [34]. All this is of vital importance due to the multiple cell cycle alterations found in AD commented on above [113,116]. In this regard, EGCG is capable of travelling across the blood–brain barrier and interferes with mitogenic signalling at the brain level, preventing the progression of an altered cell cycle in the presence of excess iron [34]. In addition, EGCG can directly potentiate the expression of p21 and p27, while diminishing the expression of D1 cyclin and pRB, abolishing re-entry to the cell cycle thanks to a primary antiproliferative action [27,116,126] (Figure 4 (10–14)). 

It is worth mentioning that the antioxidant and iron chelating activity of EGCG may be useful to inhibit ferroptosis cell death that, as mentioned before, is increased in an AD brain. In fact, there is evidence that supports that the inhibition of brain ferroptosis protects from a brain haemorrhage [127]. Additionally, it has been shown that EGCG was able to inhibit ferroptosis after spinal cord injury through protein kinase D1 phosphorylation [128]. Therefore, several authors have suggested a potential protective role of EGCG by preventing ferroptosis in Alzheimer’s disease as a therapeutic strategy [98,99] (Figure 2 (14,18)).

### 4.3. Modulating Effect of EGCG in Cell Signalling, Survival and Death Pathways

As broadly commented on in previous sections, there are multiple intracellular signalling pathways related to neuroprotection and cell survival in which Alzheimer’s disease and oxidative stress mechanisms can converge, including protein kinase C, mitogen-activated protein kinases and phosphoinositide 3-kinase pathways [119]. All of them are related to several neuronal functions, such as plasticity, synaptic morphology, and protein synthesis, which can in turn affect memory and neurodegeneration [67,119,129]. 

Briefly, EGCG has been reported to exert direct neuroprotective effects through the modulation of cell survival and death, activating ERK, Akt/PKB, PI3K and PKC pathways that improve cell survival, while inhibiting p38 and JNK ones, thus avoiding apoptosis [94,129].

#### 4.3.1. EGCG and the PKC Pathway: Implications in Alzheimer’s Disease

Protein kinase C constitutes a family of kinases whose function is phosphorylating serine/threonine residues of proteins, regulating their biological functions. Kinases from the PKC family are involved in the brain signalling network through the regulation of cell signalling, cell growth, differentiation, and apoptosis, with direct consequences on tumorigenesis, synaptic function, behaviour and cognition [130,131].

At least 12 isoforms can be found in mammals, classified in three subfamilies, according to their structure and specific requirements of the second messengers: classical PKC (cPKC), with isoforms α, βI, βII and γ; atypical PKC (aPKC), composed of isoforms ι, λ and ζ; and novel PKC (nPKC), which includes isoforms δ, ε, η, μ and θ [132]. Among these, α, γ, ε and ζ have been linked to signalling processes related to memory mechanisms and memory deficits, which has earned them the nickname of memory kinases [132]. PKC activators such as arachidonic acid, aplysiatoxins or bryostatins can improve memory [133,134] and restore synaptic and network functions [135,136], exerting anti-dementia effects [137,138]. In fact, PKC activation by arachidonic acid is one of the main mechanisms of astrocyte-induced synaptogenesis [139]. 

From a broad perspective, PKC activation has a wide range of biological effects. On synaptic transmission, for instance, it enhances the synthesis, vesicle replenishment and liberation of cholinergic, dopaminergic, glutaminergic and GABAergic neurotransmitters [140,141,142]. Relatedly, PKC has also been linked to synaptic plasticity, promoting long-term potentiation (LTP) phenomena, with PKCζ having proven to be necessary and sufficient to maintain hippocampal LTP [143] and having been associated with long-term memory [144,145], while PKCα, PKCγ and PKCε have been related to memory and learning processes [132] (Figure 5 (2–5,8–12)).

In addition, different PKC isoforms have been reported to perform opposite functions regarding cell growth, differentiation and apoptosis. Specifically, PKCθ and PKCδ promote apoptosis, while PKCα, PKCβ, PKCε and PKCζ can avoid it and promote neurite growth instead [146,147]. In addition, PKCε has been proven to directly enhance the expression of brain-derived neurotrophic factor (BDNF), which can activate complex signalling pathways, resulting in the repair of the synaptic structure and function and production of new brain cells [132,148] (Figure 5 (6–12)).

Alterations in PKC signalling pathways have been found to contribute to Alzheimer’s disease pathogenesis and are associated with memory deficits and learning difficulties [132,149], possibly establishing reciprocal interactions with pathogenic mechanisms, since PKC isoforms are also sensitive to AD-related stress factors and amyloid plaques [149]. This can be linked to the anti-dementia effects mentioned for PKC activation [137]. In the context of AD, PKC activation has been proposed to stimulate LTP and cognitive improvement, by helping reduce the amyloid load [149,150]. On top of that, PKC activation also inhibits glycogen synthase kinase 3 (GSK3), the main kinase involved in tau phosphorylation, providing yet another confluence between PKC pathways and AD pathology [151] (Figure 5 (23,24,14)).

In turn, Aβ was shown to decrease PKC levels by directly binding PKC isoforms, effectively reducing their phosphorylation and translocation, blocking their activation and inducing their degradation [152]. Aβ peptides and oligomers also inhibit RACK and intracellular receptors required in PKC activation, and actively block BDNF, which specifically links it to PKCε [149,150,152]. Indeed, PKCε has been associated with the activation of endothelin converting enzyme 1 (ECE-1) [150], one of the main enzymes involved in amyloid β degradation and amyloid plaque reduction [153,154], rendering PKCε activation and overexpression as effective methods to reduce amyloid pathology [155,156]. Parallelly, both PKCε and PKCα act as activators of α-secretase that mediate the non-amyloidogenic processing of APP, contributing to alleviate amyloidogenic buildup and promoting the generation of sAPPα that also acts as an Aβ inhibitor and exerts neuroprotective functions [38,43,132,152]. In addition, PKCε may be able to contribute to amyloid degradation by activating circulating serine proteases that can cleave Aβ [150] (Figure 5 (15,17–22))

All these intricate pathways play a role on the neuroprotective function exerted by EGCG and other catechins. EGCG has been shown to contribute to PKC pathways through the direct activation of PKC by means of a fast phosphorylation that promotes the beneficial, neuroprotective effects previously detailed [27,149]. In vivo studies on C57BL mice under a 2 mg/kg/day consumption schedule of EGCG proved that this catechin is also able to induce a fast translocation of PKCα, which prevented PKCα depletion and counteracted the increase in the apoptotic protein Bax in neurons [34]. Synergically, EGCG has also been demonstrated to induce a rapid proteasomal degradation of Bad, another proapoptotic protein, via PKC activation [157] (Figure 5 (1,25–27)).

Moreover, the specific activation of PKCα and PKCε entails the stimulation of their anti-AD pathology pathways described above. Hence, several studies using an in vivo mice model of AD such as Tg2576 and APP_695_SWE proved that EGCG induces PKCε-mediated ECE-1 activation and promotes the non-amyloidogenic processing of APP through PKC activation, promoting sAPPα production and a significant reduction in Aβ and amyloid plaque levels in the brain [158] (Figure 5 (19–22)). 

#### 4.3.2. EGCG and the MAPK Pathway: Implications in Alzheimer’s Disease

The mitogen-activated protein kinases’ pathway constitutes another crucial signalling cascade in terms of cell proliferation, differentiation, apoptosis and survival, as well as inflammation and innate immunity. Mammal MAPK are grouped in three categories: c-Jun N-terminal kinases (JNK), with three different isoforms termed JNK1, JNK2 and JNK3; p38 kinases, with isoforms α, β, γ and δ; and extracellular signal-regulated kinases (ERK), composed of isoforms ERK1, ERK2 and ERK5 [159,160]. All of them act as transductors of extracellular stimuli by unfolding a phosphorylation cascade composed of, at least, three components: a MAPK kinase (MAP3K) that phosphorylates and activates another MAPK kinase (MAP2K), which in turn phosphorylates and activates a MAPK [159]. This activated MAPK is then able to phosphorylate several targets, including transcription factors and antiapoptotic and proapoptotic proteins [159,161].

Each individual MAPK signalling pathway is activated as a result of complex interactions between different kinase components or through a signalling complex composed of several kinases and a scaffold protein [159]. Broadly speaking, we can assert that ERK1/2-mediated pathways are activated by growth factors and stimulate cell proliferation, migration, differentiation and survival [159,162], while p38 and JNK are activated by stress factors such as oxidative stress or inflammation, and are therefore responsible for inflammatory and stress responses, autophagy and apoptosis, although they can also participate in cell differentiation [163,164,165]. 

Given the myriad of pivotal processes in which these signalling pathways take part, it should not come as a surprise that MAPK signalling is altered in multifactorial pathogenic processes such as AD and other neurodegenerative disorders. In Alzheimer’s disease, Aβ-induced oxidative stress and microglial activation have both proven to be mediators of MAPK p38 signalling [166,167,168], which promotes apoptosis but also acts as a kinase of tau protein [169], further contributing to Alzheimer’s disease (Figure 6 (9,10)). In line with this, the inhibition of IL-1β signalling during neuroinflammation in AD ameliorated tau pathology and improved cognitive function in a p38-dependent manner [170]. Other AD-related pathological changes, such as mitochondrial dysfunction and mitochondrial dynamics alteration seem to be mediated by ERK pathways, since their blockage has been shown to improve mitochondrial morphology and function and reverse alteration in the expression and distribution of mitochondrial dynamics’ proteins such as DLP1 and Mfn2 [171] (Figure 6 (8)).

As for the role of EGCG in relation to these signalling pathways, the green tea catechin was proven to preclude ERK1/2 downregulation mediated by oxidative stress, which results in increased cell survival, both in nervous and non-nervous tissue [165]. In accordance with the antioxidative effect discussed before, EGCG was also shown to induce antioxidant defence systems through the activation of the Keap1/Nrf2/ARE pathway and antioxidative enzymes through Akt and ERK1/2 activation [172,173]; albeit EGCG seemed to be unable to exert any activating effect on ERK1/2 in the absence of oxidative stress conditions [27] (Figure 6 (1–3)).

Additionally, EGCG delivers an orchestrated effect, by also inhibiting ROS-induced phosphorylation in MAPK from the JNK and p38 pathways, which rendered them inactive [165], while also inhibiting hydrogen-peroxide-dependent caspase 3 activation, thus avoiding apoptosis [165,174] (Figure 6 (6,10)).

#### 4.3.3. EGCG and the PI3K/Akt Pathway: Implications in Alzheimer’s Disease

Another signalling pathway of paramount importance for cell survival and cell cycle progression, as well as metabolism, cell motility and transcription, is the protein kinase B (PKB, also termed Akt) pathway. Mammals display three PKB isoforms: α, β and γ (Akt 1, 2 and 3) [175], which are activated by phosphoinositide 3 kinase (PI3K) to exert antiapoptotic functions [175,176] (Figure 7 (1)). 

PI3K, in turn, can be activated by a number of stimuli, including trophic factors such as nerve growth factor (NGF), insulin-like growth factor (IGF-1) or BDNF [175,177]. Upon activation, PI3K catalyses the phosphorylation of phosphatidylinositol (4,5)-bisphosphate (PIP2) to phosphatidylinositol (3,4,5)-trisphosphate (PIP3), while the inverse reaction is catalysed by phosphatase PTEN (phosphatidylinositol (3,4,5)-triphosphate 3-phosphatase) [178,179]. The latter recruits Akt and the serine/threonine kinase PDK1 (phosphoinositide-dependent kinase 1) to the plasma membrane and promotes a signalling cascade that culminates with the activation of PKB/Akt [175,178] (Figure 7 (1–3)).

The Akt pathway regulates different proteins from the Bcl-2 family, which includes proapoptotic (Bax, Bad…) and antiapoptotic (Bcl-2) effectors. Akt can directly inhibit the apoptotic Bad proteins and caspases and indirectly inhibit the proapoptotic effects of GSK3 by increasing the levels of antiapoptotic proteins such as Bcl-2, effectively blocking neuronal apoptosis [175,180]. Relatedly, Akt is also a regulator of metabolism, with one of its main functions being the inhibition of GSK3 via phosphorylation, which prompts the storage of glucose and glycogen and seems to be in itself an antiapoptotic mechanism [180] (Figure 7 (4–11)).

In the context of Alzheimer’s disease, the Akt pathway plays a fundamental role as one of the most potent inhibitors of GSK3, the main kinase that drives tau phosphorylation [151,181]. Indeed, AD pathology seems to be mediated, at least partly, by the dysregulation of this pathway that allows GSK3 overactivation, which consequently causes tau hyperphosphorylation [181,182,183] (Figure 7 (12)).

Once again, EGCG is able to induce the activation of the PI3K/Akt pathway, consequently leading to increased cell survival and apoptosis inhibition [165], which seems to be the mechanism by which it prevented oxidative-stress-mediated cytotoxicity in PC12 cells in vitro [184]. In addition, under oxidative stress conditions, EGCG inactivated the phosphatase PTEN—what would prevent PIP3 transformation into PIP2—increasing PKB/Akt activation [184]. In any case, these mechanisms converge in the activation of Akt, which blocks GSK3, effectively inhibiting the proapoptotic caspases’ route, preventing the liberation of cytochrome c by avoiding mitochondrial damage and precluding tau hyperphosphorylation [182,184] (Figure 7 (13–16)).

## 5. Pharmacokinetic Considerations of EGCG as a Therapeutic Tool

Current pharmacological treatments for Alzheimer’s disease are not able to provide an actual cure for the disease, but rather an alleviation of symptoms and a certain decrease in the rate of progression [1,5]. The last decade has witnessed an increase in interest towards dietary interventions for uncurable pathologies, including Alzheimer’s disease, with green tea’s polyphenols and EGCG as its main bioactive compound gaining some popularity recently due to its pleiotropic effects on several AD pathology mechanisms [26,30,94,185,186,187].

As it is often the case with bioactive compounds, we face several challenges when we intend to apply them with therapeutic purposes, which implies we need to assess their dosage, administration, absorption, distribution, bioavailability, biotransformation and excretion. Although EGCG has proven to act on a molecular level on several AD-related pathways, it is important to consider here that it presents several challenges regarding its potential therapeutic use.

For one, EGCG’s stability is highly dependent on temperature and pH [188], with pH being critical, since it achieves a higher stability at a pH range between 2.0 and 5.5 but becomes autoxidised at alkaline pH values, which explain its poor intestinal stability [189,190]. As for temperature, it can suffer autoxidation below 44 °C if pH conditions allow it, but epimerization needs temperatures over 44 °C [187], which should not be much of a problem in vivo but may need to be considered during storage [187,191]. Despite intestinal instability, EGCG can be absorbed via passive transcellular and paracellular diffusion due to its hydrophilic nature [187,192]. The slow speed of diffusion processes and the aforementioned instability at an alkaline pH explain its relatively high rate of intestinal retention, which renders it susceptible of being locally degraded by hydrolysis and intestinal microflora [187]. 

All this together contributes to the fact that EGCG does not enjoy a great bioavailability either, with data oscillating between less than 1% and 5% of this catechin found in the systemic circulation after the consumption of ~500 mg of tea [193,194], most likely due to the particularities of its structure, which hence pose interesting areas of improvement by modifying its structure [192]. 

Current approaches to safeguard the beneficial potential displayed by EGCG include the synthesis of a prodrug that would release EGCG upon biotransformation, by modifying its structure through common pharmaceutical chemistry approaches such as esterification, methylation or glycosylation [192,195]. More recent approaches are also considering different formulation strategies and administration routes, and devising delivery systems that can increase stability and absorption, such as nanoencapsulation or the use of lipidic or polymeric nanocarriers [196,197,198,199]. Albeit it is not the purpose of the current review to deepen into more clinical considerations of the potential applications of EGCG, it would be negligent to not have noted the practical difficulties of the therapeutic use of this catechin. For a detailed review on these considerations and the approaches that are being proposed to overcome such problems, we redirect the readers to the recent work of Mehmood and colleagues [187].

## 6. Discussion and Future Perspectives

In the present work, we have reviewed the molecular mechanisms by which EGCG, the main catechin present in green tea, exerts a neuroprotective function that is especially relevant in the context of Alzheimer’s disease and particularly prominent for amyloid-related pathology [158,198]. Indeed, we have detailed that EGCG can promote the non-amyloidogenic processing of APP, reducing the Aβ load and generating neuroprotective sAPPα, while also modulating tau phosphorylation, preventing NFT formation [22,34,94,199]. These effects were accompanied by a plethora of other synergistic mechanisms that contribute to the neuroprotective role, including metal chelation [94,113,116], antioxidative effects [63,184,200] and the promotion of molecular pathways linked to cell survival via the activation of PKC, PI3K, PKB/Akt and ERK1/2 [27,94,132,153,159,175].

However, these conclusions are only sustained on the basis of in vitro cultured cells and animal studies. While they open a promising window with regard to the effects of EGCG and other green tea polyphenols as neuroprotective agents, we need systematic studies in humans to assert the validity of these mechanisms and effects. A great number of clinical trials are being conducted at the moment, trying to address this very point, but some other correlation studies have already been carried out to try and assess possible relationships between green tea consumption and dementia or cognitive decline.

For instance, the so-called Tsurugaya project followed 1003 elderly Japanese subjects over 70 years old, who frequently consumed green tea [201]. The project comprised a survey assessing the frequency of tea consumption, grouping them accordingly (≤3 cups of tea per week, 4–6 cups per week, 1 or more cups of tea per day), as well as sociodemographic variables and a Mini Mental State Examination (MMSE) to evaluate cognitive function, which is a standardized test employed in the screening of dementias [201,202]. In total, 72.3% of the subjects consumed at least 1 cup of tea per day, observing a diminished prevalence of cognitive decline in this group with respect to those who consumed less tea. There was a strong inverse correlation between green tea consumption and cognitive decline, while the correlation was weak or nonexistent for black tea, oolong tea or coffee consumption [201]. These differences could be a relevant clue to assert that the effect observed for green tea is mediated by its catechins, since their concentration drastically decreases in the other types of tea [24,28]. Some authors even venture that this, together with tea consumption data, could help explain the reduced prevalence of cognitive decline and Alzheimer’s disease in the elderly in Japan with respect to North America [203].

These results, nonetheless, should be taken with caution. While several in vitro studies have proved the potential beneficial effects of EGCG in concentration ranges oscillating between 1 and 100 μmol/L, in vivo studies using animals and some studies performed in humans have shown smaller micromolar ranges of peak plasmatic concentrations of green tea catechins [194]. This may point towards the existence of other synergistic or independent effects that may explain the inverse correlations between green tea and cognitive decline. Indeed, this would also be supported by the poor pharmacokinetic performance of EGCG and other catechins [187] that, combined with extensive degradation and biotransformation, even when administered intravenously, leads to a poor bioavailability on target tissues [187,192,194].

Indeed, the use of EGCG in a therapeutic setting poses other biopharmaceutical and pharmacokinetic challenges. For instance, extensive intestinal and hepatic metabolism has been described for green tea catechins by means of phase II enzymes such as catechol-o-methyltransferases, sulfotransferases or glucuronyltransferases [194,204], and by resident intestinal microflora [187,194]. Among the metabolites that can be found in humans after tea consumption are glucuronic, sulfate and methylated conjugates, as well as microflora-mediated ring fission and phenolic acid catabolites, several of which can exert their own biological functions, depending on their tissular distribution [194]. Despite intestinal instability and the described metabolism that catechins may undergo, the intestine has proven to be the main route of absorption of catechins by passive diffusion [187]. In fact, in a study carried out in ileostomy patients after green tea consumption, 70% of catechins were detected in ileal fluid within 24 h of consumption, with plasmatic concentrations that barely reached 100–250 nM (around 10–25% of the minimal effective concentration in in vitro studies) [205].

EGCG metabolites have also been proven to cross the BBB and induce active effects, such as neuritogenesis [31], which, together with the poor bioavailability of EGCG, suggests that these metabolites may contribute to the potential beneficial effects of green tea consumption described above. Namely, EGCG suffers a hydrolyzation process that separates gallic acid from epigallocatechin (EGC), with the latter being subjected to further biotransformation that results in at least 11 ring-fission metabolites, with 5-(3,5-dihydroxyphenyl)-γ-valerolactone and 5-(3,4,5-trihydroxyphenyl)-γ-valerolactone being the most prominent in human plasma, urine and bile [31,206]. In addition, these metabolites have also been reported to exert antioxidative, anti-inflammatory and immunomodulatory activities, among others [31]. Thus, it is more than likely that the effects observed upon green tea consumption are related to both EGCG and its metabolites, sequentially (first EGCG and, upon biotransformation, its metabolites) or via synergic effects between all of them. Readers are encouraged to consult the review by Pervin et al. [31] for more in-depth insights into the bioavailability and bioactivity of EGCG metabolites.

If, indeed, the observed effect arises from both EGCG and its metabolites, the establishment of a dosing regimen turns into a rather complicated matter, since current evidence does not allow us to ascertain which specific compounds play a part in the overall effect and what their specific contribution is. Since there is no evidence of neuroprotective effects of the aforementioned metabolites of EGCG in vivo [31], we could assume that EGCG is the main effector, but we would still have a problem of bioavailability, which would lead to increasing EGCG dosing to avoid a low brain concentration. However, the use of higher EGCG concentrations to palliate such a problem is not recommended, since it can exert toxic functions via a bimodal effect. In low concentrations, green tea polyphenols act as antioxidants by diminishing ROS and activating antioxidative enzymes, but higher doses can result in ROS production that would promote oxidative stress [187,194]. Additionally, the consumption of EGCG by means of green tea consumption would require the ingestion of 8–16 cups of tea per day to solve this bioavailability problem, which really limits its implementation as part of a dietary intervention [194]. Even if the ingestion of such amounts of green tea was feasible, it is important to bear in mind that the potential effects of dietary interventions do not arise from single compounds or ingredients within food and beverages, but rather from the overall effect obtained from synergistic and antagonistic effects of the compounds within each product in the context of a specific dietary pattern [17]. This can help explain the fact that studies such as the Tsurugaya project have found correlations between green tea consumption and the avoidance of cognitive decline even with fewer cups of tea per day than the estimate mentioned in this paragraph [194].

A rapidly growing field of research is now focusing on nanotechnology to encapsulate bioactive compounds in an effort to provide a solution to these bioavailability and instability problems associated with green tea catechins, which are currently classified as class III compounds in the biopharmaceutical classification system (high solubility, low permeability) [197,204]. Reducing the size of the formulations and delivery systems to the nanoscale allows us to vastly increase the surface area to volume ratio, which can result in the enhancement of bioactivity [204]. In this regard, different methods are being tested at the moment regarding the improvement of the biopharmaceutical characteristics of EGCG, including nanoencapsulation and the use of nanocarriers [187,204]. Current studies have already proven the effectiveness of these approaches [187,198].

Finally, the fact that we cannot fully explain the neuroprotective effects of EGCG and other green tea catechins in humans due to the lack of effective plasmatic concentrations does not invalidate the mechanistic considerations discussed throughout this review. In fact, it is possible that the beneficial effects arise from synergistic interactions between diverse green tea catechins and other components of green tea or other dietary and pharmacological compounds. Taking this into account, other studies should also focus on potential pharmacological interaction by means of crossed pharmacokinetic studies, which are already being carried out for certain, rather frequent drugs [207,208].

In any case, the pleiotropic effects described for EGCG during this work and for other green tea components in the pathogenic mechanisms associated with Alzheimer’s disease pose a promising research avenue to explore, especially in the context of potential dietary interventions on an as-of-yet incurable disease.

## Figures and Tables

**Figure 1 antioxidants-12-01460-f001:**
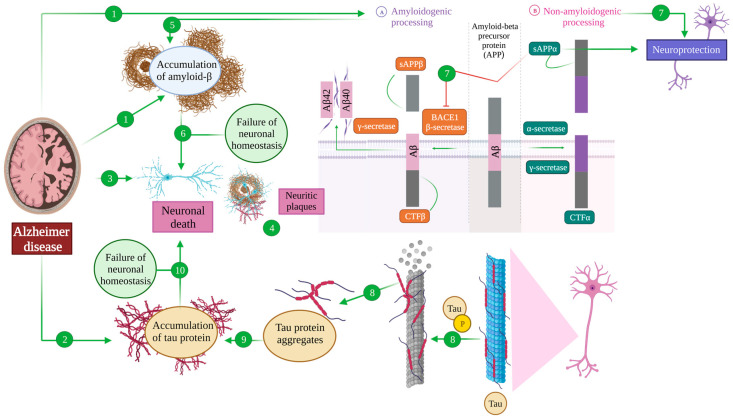
Amyloid β and tau pathology in Alzheimer’s disease. The main hallmarks of an AD brain are the accumulation of extracellular amyloid-β senile plates (1), the accumulation of intracellular neurofibrillary tangles of tau (2) and neuronal death (3). The areas in which these three features coexist are called neuritic plates (4). Amyloid precursor protein (APP) processing can take place through the amyloidogenic pathway (A), being cleaved by β-secretase (BACE), generating the sAPPβ and CTFβ fragments. The latter is subsequently processed by γ-secretase, generating Aβ42 and Aβ40, which induce amyloid β pathology in an AD brain (5) that ultimately affects neuronal homeostasis and induces cell death (6). On the contrary, in the non-amyloidogenic pathway (B), APP is cleaved by α- and γ-secretases, generating CTFα and sAPPα fragments that exert a neuroprotective function because a diminished APP conversion to Aβ and sAPPα inhibits BACE (7). On the other hand, tau is a microtubule-associated protein that establishes their structure. Post-translational modifications of tau, including phosphorylation, induce the detachment of tau and microtubule destabilization (8) and lead to tau aggregation (9). The accumulation of tau protein causes a failure of neuronal homeostasis whose final consequence is neuronal death (10).

**Figure 2 antioxidants-12-01460-f002:**
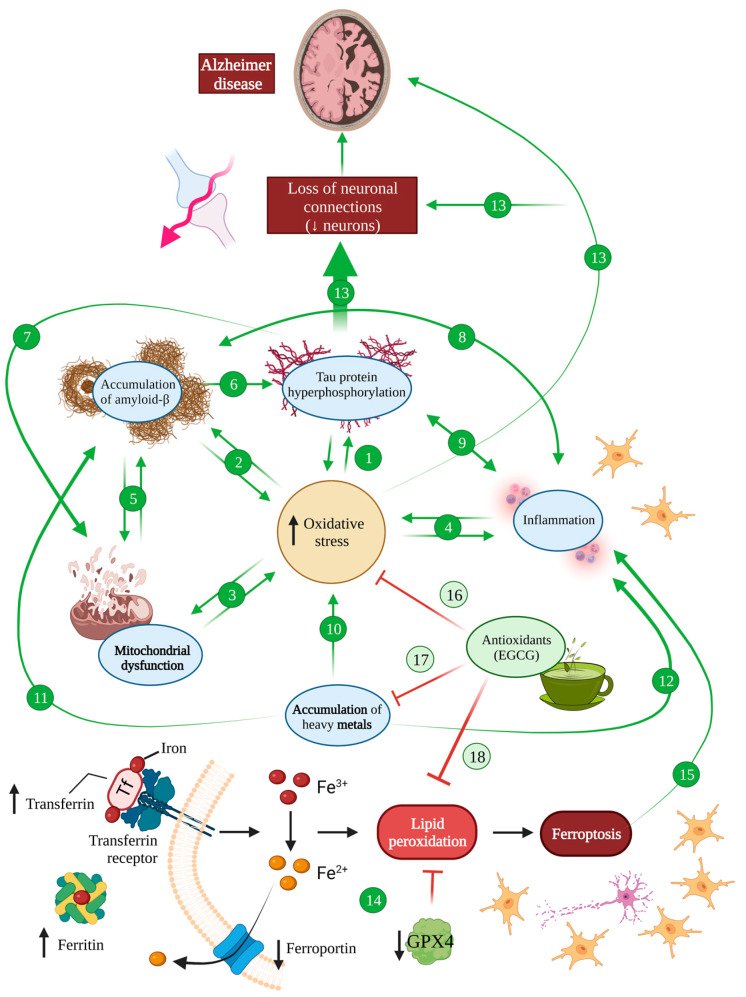
Oxidative stress in Alzheimer’s disease. There is an increase in oxidative stress in an AD brain, which induces hyperphosphorylated tau (1) and amyloid-β (2) accumulation, mitochondrial dysfunction (3) and inflammation (4). At the same time, these processes also induce oxidative stress, generating a vicious cycle. Mitochondrial dysfunction is related to Aβ accumulation and, at the same time, Aβ is able to impair mitochondrial function (5). The accumulation of Aβ also induces tau hyperphosphorylation (6), which provokes mitochondrial dysfunction (7). Either Aβ (8) or tau (9) accumulation induce inflammation, which, in the same way, favours the accumulation of Aβ and tau. The accumulation of metals is one of the main causes of ROS increasement (10) in an AD brain and it also favours Aβ pathology (11) and inflammation processes (12). All these features together induce the loss of neuronal connections and the reduction in neurons shown in an AD brain (13). Ferroptosis is increased in an AD brain due to increased levels of iron, ferririn and transferrin and diminished ferroporter and GPX4 (14) that causes neuron necroptotic death and subsequent inflammation (15). EGCG is able to block oxidative stress (16) and, thanks to its chelating activity, it counteracts the accumulation of heavy metals (17) and ferroptosis (18), protecting the brain from all these harmful effects.

**Figure 3 antioxidants-12-01460-f003:**
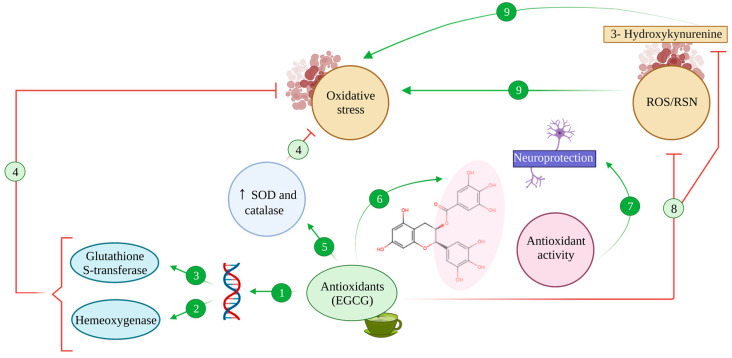
Antioxidant activity of EGCG. The antioxidant compound EGCG binds to antioxidant regulatory elements (ARE), inducing the expression of stress response genes (1) such as haem oxygenase (2) and glutathione-S-transferase (3) enzymes, counteracting oxidative stress processes (4). Additionally, EGCG enhances the activity of SOD and catalase (5), which reduce oxidative stress (4). Thanks to its hydroxyl groups, EGCG harbours chelating properties (6) that exert antioxidant activity, promoting neuroprotective effects (7). Also, EGCG can inhibit the production of ROS/RNS and 3-Hydroxykynurenine effect due to this antioxidant capacity (8), avoiding oxidative stress (9).

**Figure 4 antioxidants-12-01460-f004:**
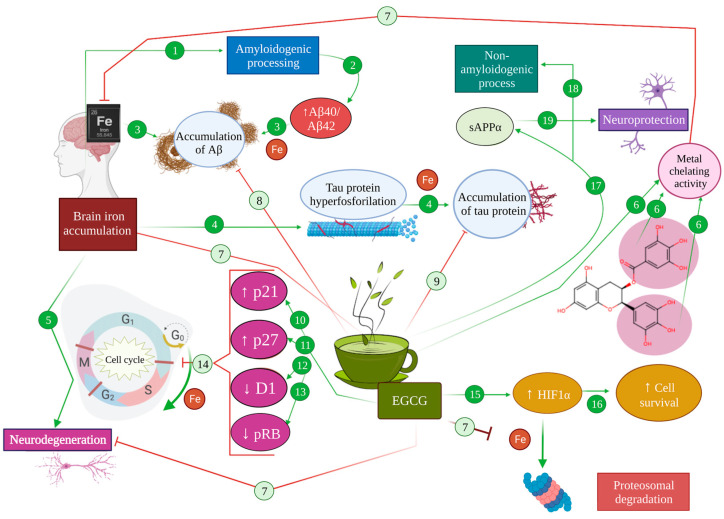
Effect of metal chelation mediated by EGCG in Alzheimer’s pathology. In an AD brain, iron accumulation promotes amyloidogenic processing (1), increasing Aβ40/Aβ42 levels (2), leading to the accumulation of Aβ (3). Moreover, brain iron accumulation produces an increase in tau hyperphosphorylation (4) that results in microtubule destabilization and the accumulation of tau protein (5). As a consequence, iron produces neurodegeneration (6). The metal chelator activity of EGCG (6) inhibits iron accumulation in the brain (7), therefore diminishing the accumulation of Aβ (8) and tau protein (9). Moreover, EGCG can directly potentiate the expression of p21 (10) and p27 (11), while diminishing the expression of cyclin D1 (12) and pRB (13), abolishing cell cycle re-entry (14). On the other hand, EGCG can promote the activation of HIF-1α (15), inducing the expression of cell survival genes (16). EGCG promotes the production of SAPPα (17) and the non-amyloidogenic processing of APP (18), generating neuroprotection effects (19).

**Figure 5 antioxidants-12-01460-f005:**
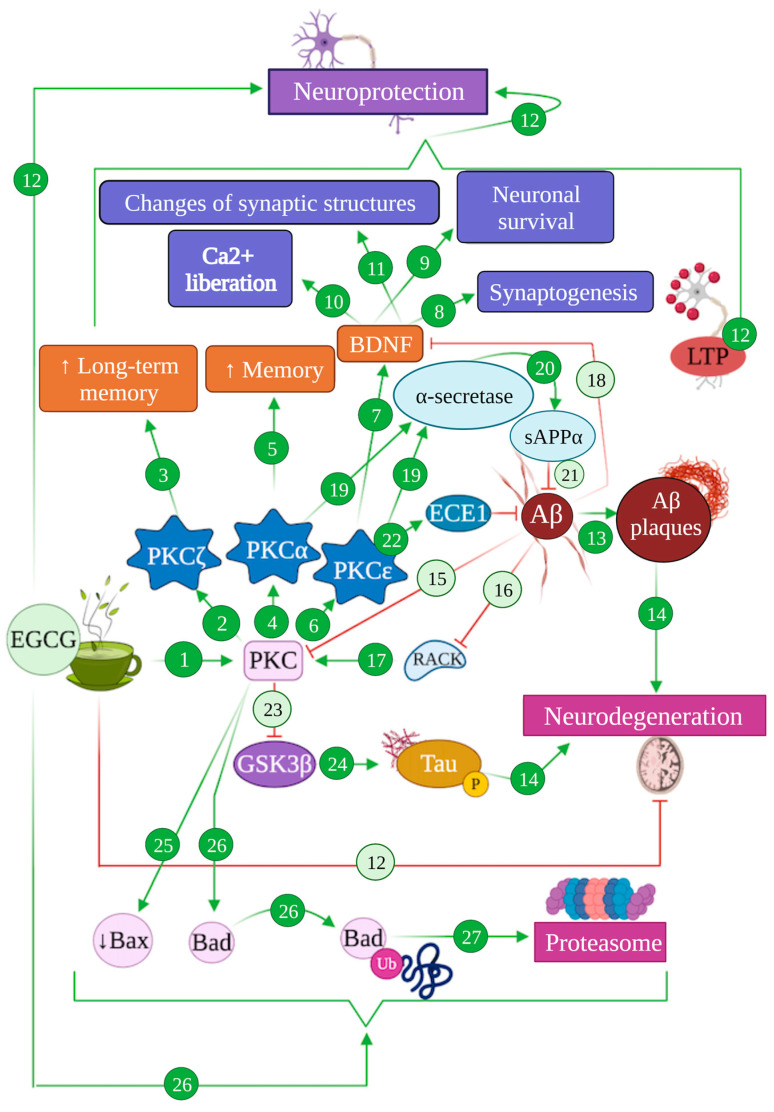
Modulation of PKC-mediated pathways by EGCG in Alzheimer’s disease. EGCG can induce the activation of PKC pathways (1), specifically the isoform PKCζ (2) that enhances long-term memory (3), PKCα (4) that results in an increase in memory function (5) and PKCε (6) that activates BDNF factor (7), promoting synaptogenesis (8), neuronal survival (9), Ca2+ liberation (10) and changes of synaptic structures (11), altogether promoting neuroprotection (12). The generation of Aβ produces Aβ plaques (13), promoting neurodegeneration processes (14). Aβ also inhibits the PKC pathway (15) and RACK (16), whose receptors are required to activate PKC (17), and blocks BDNF (18). Moreover, the isoforms PKCα and PKCε activate α-secretase (19), promoting the generation of sAPPα (20) that inhibits Aβ production (21). PKCε activates ECE1 (22), which degrades Aβ (21). The activation of the PKC pathway inhibits GSK3β (23), which is involved in tau hyperphosphorylation (24) that finally leads to neurodegeneration processes (14). EGCG produces the reduction in protein Bax (25) and promotes the degradation of protein Bad (26) through the proteasomal system (27).

**Figure 6 antioxidants-12-01460-f006:**
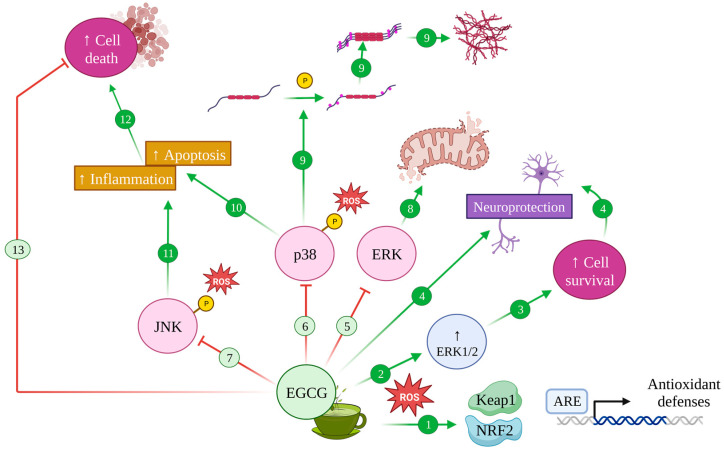
Modulation of MAPK pathway by EGCG in Alzheimer’s disease. Upon oxidative stress, EGCG induces antioxidant defences with the activation of the Keap1/Nrf2/ARE pathway (1) and increases ERK1/2 (2), promoting cell survival (3) and neuroprotective effects (4). Conversely, EGCG inhibits the ERK (5), p38 (6) and JNK (7) pathways whose effects contribute to mitochondrial disfunction and altered dynamics (8), induce tau protein hyperphosphorylation and aggregation (9) and increase apoptosis (10) and inflammation (11), leading to cell death (12). Therefore, EGCG can avoid this cell death (13).

**Figure 7 antioxidants-12-01460-f007:**
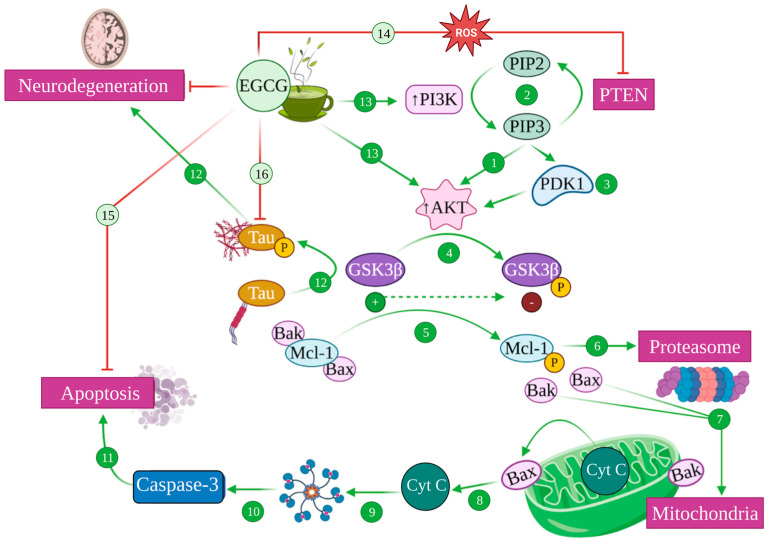
Modulation of PI3K/Akt pathway mediated by EGCG. Akt/protein kinase B (PKB) is activated by Phosphatidyl Inositol 3 Kinase (PI3K) (1), which catalyses the conversion of phosphatidyl inositol (4,5) biphosphate (PIP2) into phosphatidyl inositol (3,4,5) triphosphate (PIP3), a process that is reversed by phosphatidylinositol (3,4,5)-triphosphate 3-phosphatase (PTEM) (2). PIP3 also recruits phosphoinositide-dependent kinase 1 (PDK1) to the plasma membrane, activating Akt (3). Akt phosphorylates glycogen synthase kinase 3 (GSK3), inhibiting its function (4). Active GSK3 is able to phosphorylate Mcl-1 (5), which is targeted for proteasomal degradation (6), liberating Bax and Bak proapoptotic factors (7). This causes the permeabilization of the mitochondrial outer membrane, releasing cytochrome c (Cyt C) (8) that attaches to Apaf-1, generating the apoptosome (9), leading to the activation of caspase 3 (10) that induces apoptosis (11). GSK3 also phosphorylates tau, inducing its aggregation and subsequent neurodegeneration (12). EGCG can ultimately increase PI3K and Akt activity (13) and inhibit PTEM in the presence of ROS (14), inhibiting apoptosis (15), phospho-Tau aggregation (16) and, therefore, generating neuroprotection.

## Data Availability

Data is available in the original articles cited in the present review.

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
