# Peer review of "Alzheimer’s Disease and Green Tea: Epigallocatechin-3-Gallate as a Modulator of Inflammation and Oxidative Stress"

_antioxidants, 2023, doi:10.3390/antiox12071460_

Round 1

Reviewer 1 Report

Dear authors,

Authors reviewed on the effects of epigallocatechin-3-gallate in green tea for potential benefits for Alzheimer's disease.

Authors covered various pathways on the mode of actions of epigallocatechin-3-gallate.

The 1st important huddle for any AD therapeutics is crossing of BBB.

Epigallocatechin-3-gallate was shown that it would not cross BBB.

Then, would epigallocatechin-3-gallate benefit in AD?

If authors still think epigallocatechin-3-gallate would be beneficial for AD, please discuss the mode of action of epigallocatechin-3-gallate outside of BBB.

Authors need to discuss the metabolites of epigallocatechin-3-gallate upon intake into our body and their clearances.

Authors should also calculate the amount of tea or epigallocatechin-3-gallate to be taken by AD patients to be effective (effective dosage)

English needs to be improved, where the facts are in present tense, and the referenced statements should be in the past tense.

Author Response

Dear Reviewer #1,

Please, find enclosed the revised version of our manuscript 2452499, entitled Alzheimer's disease and green tea: epigallocatechin-3-gallate as a modulator of inflammation and oxidative stress; by Victor Valverde-Salazar, Daniel Ruiz-Gabarre and Vega García-Escudero.

By following your indications, we have prepared this revised version taking into account both reviewers’ input and address all of the issues raised by you point by point in the present letter:

Reviewer #1

Dear authors,

Authors reviewed on the effects of epigallocatechin-3-gallate in green tea for potential benefits for Alzheimer's disease. Authors covered various pathways on the mode of actions of epigallocatechin-3-gallate.

The 1st important huddle for any AD therapeutics is crossing of BBB. Epigallocatechin-3-gallate was shown that it would not cross BBB. Then, would epigallocatechin-3-gallate benefit in AD? If authors still think epigallocatechin-3-gallate would be beneficial for AD, please discuss the mode of action of epigallocatechin-3-gallate outside of BBB.

We want to thank the author for pointing this out, since it makes us realize maybe we did not state it clearly enough on our original manuscript. However, it is important to highlight that, contrary to the reviewer idea, EGCG is able to cross the blood-brain barrier. In our original text we wrote in the last paragraphs of introduction:

The effects of EGCG and its abundance within green tea, together with the fact that it can be efficiently absorbed in the intestine [28,29] constitute two key points of the potential therapeutic use of EGCG. In addition, since AD pathophysiology occurs mainly in the brain, the ability to cross the blood-brain barrier is a necessary feature of any potential therapeutic agent. In this regard, EGCG has been proven to cross the blood-brain barrier even at very low concentrations [30-33], which positions this catechin as a potential mediator with beneficial properties for Alzheimer’s disease and other forms of neurodegeneration [22,26,27,34].

Several other authors researching diverse matters including cancer or Down’s syndrome have reported as well that EGCG can indeed cross the blood-brain barrier, even at very low levels. Some references to this point (in addition to the one already mentioned in the text) are:

  • Wyganowska-Świątkowska M, Matthews-Kozanecka M, Matthews-Brzozowska T, Skrzypczak-Jankun E, Jankun J. Can EGCG Alleviate Symptoms of Down Syndrome by Altering Proteolytic Activity? Int J Mol Sci. 2018 Jan 15;19(1):248. org/10.3390/ijms19010248
  • Pervin, M.; Unno, K.; Takagaki, A.; Isemura, M.; Nakamura, Y. Function of Green Tea Catechins in the Brain: Epigallocatechin Gallate and its Metabolites. Int. J. Mol. Sci. 2019, 20, 3630. https://doi.org/10.3390/ijms20153630
  • Unno, K., Pervin, M., Nakagawa, A., Iguchi, K., Hara, A., Takagaki, A., ... & Nakamura, Y. (2017). Blood–Brain Barrier Permeability of Green Tea Catechin Metabolites and their Neuritogenic Activity in Human Neuroblastoma SH‐SY5Y Cells. Molecular nutrition & food research, 61(12), 1700294. https://doi.org/10.1002/mnfr.201700294

However, to highlight this point so as to make perfectly clear EGCG’s ability to cross the BBB, we have rephrased this part and added extra references, as follows:

The effects of EGCG and its abundance within green tea, together with the fact that it can be efficiently absorbed in the intestine [28,29] constitute two key points of the potential therapeutic use of EGCG. In addition, since AD pathophysiology occurs mainly in the brain, the ability to cross the blood-brain barrier is a necessary feature of any potential therapeutic agent. In this regard, EGCG has been proven to cross the blood-brain barrier even at very low concentrations [30-33], which positions this catechin as a potential mediator with beneficial properties for Alzheimer’s disease and other forms of neurodegeneration [22,26,27,34].

Authors need to discuss the metabolites of epigallocatechin-3-gallate upon intake into our body and their clearances.

It is not the intention of this review to focus on EGCG metabolites and their potential effects, but rather, to agglutinate in one work the proposed pathways by which EGCG may exert neuroprotective functions that may help tackle the molecular pathogenic mechanisms of Alzheimer’s disease.

Nonetheless, we want to thank the reviewer because we think adding the idea of EGCG’s metabolites to our discussion truly improves our work and thus, have decided to delve a little deeper and have added a specific paragraph in the Discussion and future perspectives section addressing these metabolites. Additionally, several references to the potential synergic or sequential effects of EGCG and its metabolites have been added throughout the discussion.

Authors should also calculate the amount of tea or epigallocatechin-3-gallate to be taken by AD patients to be effective (effective dosage)

Thanks to the reviewer’s suggestion to include the action of EGCG metabolites in the text, we want to ascertain that establishing a specific dosing regimen is not that straightforward, since we cannot ensure that the neuroprotective effects are exerted exclusively by EGCG or, if resulting from synergic effects with its metabolites, which ones play a part and to which extent. In any case, we have added a paragraph to this matter in Discussion and future perspectives.

In addition, as stated in the introduction, we aimed to review the evidence of EGCG’s molecular effects that can help tackle a disease where the lack of effective treatment has prompted a shift towards alternative therapeutic options, such as dietary interventions. In this context, we discuss throughout the Discussion and perspectives section the amount of green tea cups that have to be consumed in order to exert a neuroprotective effect and we contrast it with previous results that show an inverse correlation between green tea consumption and cognitive decline.

Due to the reasons stated above and discussed in the text, and since we are not proposing the use of EGCG supplementation, but rather its inclusion within dietary interventions, it is not our objective to establish a specific dosage for this particular compound, but trust that future clinical studies will help establish such thresholds.

Comments on the Quality of English Language

English needs to be improved, where the facts are in present tense, and the referenced statements should be in the past tense.

English usage has been reviewed throughout the whole text and verbs’ tenses have been modified where appropriate.

We would like to sincerely thank you for your time and your accurate suggestions that have undoubtedly improved the quality of the presented review by adding relevant information and improving clarity. We trust that all reviewers’ concerns have been addressed and the manuscript is now suitable for publication in Antioxidants.

Yours sincerely,

Vega García-Escudero

Reviewer 2 Report

Review on the manuscript of Valverde-Salazar, V. et al.: “Alzheimer's disease and green tea: epigallocatechin-3-gallate as a modulator of inflammation and oxidative stress”.

In this manuscript, the authors review the available data on the neuroprotective and neuroregenerative role of Epigallocatechin gallate (EGCG) and the molecular mechanisms involved in such effects, particularly in Alzheimer’s disease.

The manuscript is very clear and well written. Thus, the issues that arise to me are listed below, so, I hope the authors find the following comments and suggestions useful.

1 - Ferroptosis represents another mechanism that has been increasingly linked to Alzheimer’s disease. Several studies have demonstrated that EGCG protects against ferroptosis in different organs. Thus, I recommend authors to discuss ferroptosis in the manuscript and explore how ferroptosis modulation by EGCG could be beneficial for Alzheimer’s disease.

2 – Figures are good, but it is difficult to understand what they mean. Thus, I recommend authors to add numbers to the different steps illustrated in the figures and provide a more detailed description of these steps in the figure legend.

3 – The discussion section is mainly describing future perspectives. Thus, I recommend authors make discussion and futures perspectives as two independent sections.

Author Response

Dear Reviewer #2,

Please, find enclosed the revised version of our manuscript 2452499, entitled Alzheimer's disease and green tea: epigallocatechin-3-gallate as a modulator of inflammation and oxidative stress; by Victor Valverde-Salazar, Daniel Ruiz-Gabarre and Vega García-Escudero.

By following your indications, we have prepared this revised version taking into account both reviewers’ input and address all of the issues raised by you point by point in the present letter:

Reviewer #2

Review on the manuscript of Valverde-Salazar, V. et al.: “Alzheimer's disease and green tea: epigallocatechin-3-gallate as a modulator of inflammation and oxidative stress”.

In this manuscript, the authors review the available data on the neuroprotective and neuroregenerative role of Epigallocatechin gallate (EGCG) and the molecular mechanisms involved in such effects, particularly in Alzheimer’s disease.

The manuscript is very clear and well written. Thus, the issues that arise to me are listed below, so, I hope the authors find the following comments and suggestions useful.

1 - Ferroptosis represents another mechanism that has been increasingly linked to Alzheimer’s disease. Several studies have demonstrated that EGCG protects against ferroptosis in different organs. Thus, I recommend authors to discuss ferroptosis in the manuscript and explore how ferroptosis modulation by EGCG could be beneficial for Alzheimer’s disease.

We want to kindly acknowledge the reviewer for highlighting this important point. Several paragraphs have been included regarding ferroptosis in the text.

In the Alzheimer's disease and oxidative stress section we have addressed the relevance of ferroptosis in the disease in the following paragraph:

Moreover, ferroptosis has been described as a type of programmed cell death caused by iron overload that leads to the failure of the glutathione-dependent antioxidant defenses, resulting in the ROS overproduction and accumulation of lipid peroxides. During ferroptosis neurons release lipid metabolites from inside the body which are harmful to the surrounding neurons causing inflammation, having significant implications in haemorrhagic stroke, Alzheimer's disease and Parkinson’s disease [97] (Figure 2). It has been suggested that cellular senescence in AD involving astrocytes, oligodendrocytes, microglia and even neurons increase iron content potentially leading to neuronal cell death by ferroptosis and subsequent inflammation processes [98]. Indeed, magnetic resonance imaging has shown iron deposits in AD brain as well as increased transferrin and ferritin suggesting an augmented iron intake in these patients [99]. At the same time, the transporter that mediates the secretion of Fe3+ named ferroportin has been shown to be downregulated by Aβ diminishing iron excretion [99,100]. Additionally, Aβ has been shown to decrease the levels of GPX4, an antioxidant enzyme that regenerates reduced glutathione inhibiting ferroptosis [99]. Therefore, AD brain is very susceptible to develop ferroptosis pointing out the inhibition of this type of cell death as therapeutic strategy for AD [98,99,101].

In the Iron-chelating effects of EGCG section we have discuss the possible role of EGCG in avoiding ferroptosis in the following paragraph:

It worth mentioning that antioxidant and iron chelating activity of EGCG may be useful to inhibit ferroptosis cell death that, as mentioned before, is increased in AD brain. In fact, there is evidence that supports that inhibition of brain ferroptosis protects form haemorrhagic brain [127]. Additionally, it has been shown that EGCG was able to inhibit ferroptosis after spinal cord injury through Protein Kinase D1 phosphorylation [128]. Therefore, several authors have suggested a potential protective role of EGCG by preventing ferroptosis in Alzheimer’s disease as therapeutic strategy [98,99] (Figure 2 (14, 18)).

Due to the relevance of this point the term ferroptosis has been included also in the abstract in the following sentence:

Specifically, EGCG acts as an antioxidant by regulating inflammatory processes involved in neurodegeneration such as ferroptosis, microglia-induced cytotoxicity and by inducing signalling pathways related to neuronal survival.

2 – Figures are good, but it is difficult to understand what they mean. Thus, I recommend authors to add numbers to the different steps illustrated in the figures and provide a more detailed description of these steps in the figure legend.

According to reviewer’s suggestion we have include number to each process represented in the figures and specific explanation of these steps in the corresponding figure legends. Additionally, we have included these numbers in the reference to the figures through the text. We trust that this will help the reader to easily follow the figures.

3 – The discussion section is mainly describing future perspectives. Thus, I recommend authors make discussion and futures perspectives as two independent sections.

Once again, we thank the reviewer for their input. However, we have added more information to the discussion and feel that it is more informative to intertwine the discussion of previous evidence with the future perspectives described. Thus, we have decided not to separate these in two sections but rather rename the section to Discussion and future perspectives to acknowledge the point made here by the reviewer.

We would like to sincerely thank you for your time and your accurate suggestions that have undoubtedly improved the quality of the presented review by adding relevant information and improving clarity. We trust that all reviewers’ concerns have been addressed and the manuscript is now suitable for publication in Antioxidants.

Yours sincerely,

Vega García-Escudero

Round 2

Reviewer 1 Report

Dear Authors,

Authors revised the previous manuscript accordingly to the comments.

I recommend this manuscript to be accepted with minor revisions.

Still, there are many mistakes with verb tenses.